# Influence of farmland confirmation on farmland abandonment in China

**Zhidong Wu[1], Wolin Zheng[2]\*, Zechen Yang[1]**

1 School of Economics and Management, South China Agricultural University, Guang Zhou, China, 2 School of Credit Management, Guangdong University of Finance, Guang Zhou, China

☯ These authors contributed equally to this work.
\* zhengwolin@163.com

**Data Availability Statement:** China Labor-Force Dynamic Survey (CLDS) is a publicly available dataset. When applying for the right to use the data, the data user should provide true personal information to the Social Science Research Center of Sun Yat-sen University (hereinafter referred to

## Abstract

The general view is that land ownership affirmation provides incentives for farmers to internalize external benefits, optimizes farmers' allocation of agricultural production factors, and then reduces farmers' farmland wastage behavior. This study examines the influence of residual control and claim rights in farmland right confirmation on farmers' farmland wastage behavior. Results show that residual control rights guarantee the farmers' exclusive right to use the farmland independently, and residual claim stimulates the farmers to pursue the goal of agricultural production surplus value. However, the residual claim rights are related to the constraint conditions of agricultural production; thus, the farmland right confirmation is situational dependent on farmers' farmland wastage behavior. The surplus value of the farming output of low-income families is low, and the willingness to realize the surplus claim through agricultural reproduction is weak. Residual control reduces the risk of land loss, accelerates the transfer of the labor force, and shows the behavior of farmland wastage. Nonpoor households with high agricultural production surplus value tend to increase the allocation of agrarian production factors to maximize the income, improve the allocation efficiency of agricultural land resources, and reduce farmland wastage behavior. Conclusion: The implementation effect of accurate farmland affirmation is progressive and internally unbalanced. The institutional basis of matching policy should be to deal with the relationship between residual control right and residual claim right.

## 1. Introduction

At the Central Economic Work Conference in 2016, the strategy of Storing Grain on the Ground and Storing Grain on Technology was proposed to accelerate the implementation of the system [1]. The report to the 19th National Congress of the Communist Party of China in 2017 pointed out the need to "ensure national food security and secure the rice bowls of the Chinese people in our own hands." The No. 1 document of the central government in 2019 stated the new requirements of "consolidating the agricultural foundation and ensuring the effective supply of important agricultural products." The cultivated land resource is an essential material input of grain production and the core of implementing grain security strategy. The nature of the agricultural policy is allocating farmland resources effectively, improving

as the "Data Provider") and submit the completed "Basic Information of Data User Applicant Form" to the "Data Provider" via csyjzxsys@163.com, by fax or by mail to the "Data Provider".

**Funding:** This work was supported by National Social Science Foundation of China (No.20&ZD170); and the Major projects of National Social Science Found of China (21&ZD090); and Academic year 2019-2020 Doctoral Dissertation Scholarship of Tsinghua Rural Studies (No.201901). The funders had no role in study design, data collection and analysis, decision to publish, or preparation of the manuscript.

**Competing interests:** The authors have declared that no competing interests exist.

land productivity, and realizing Pareto improvement in land resource allocation [2]. On the contrary, the phenomenon of idle farmland caused by the large-scale transfer of rural labor force exists widely. China's agricultural land use evolves from the *Agricultural Expansion Stage* to the *Agricultural Contraction Stage* [3], and the phenomenon low efficiency of land resource utilization in China is relatively apparent [4]. Among them, the western region shows the loss of the efficiency of land resource allocation due to the *Non-agricultural Land Conversion*. On the contrary, the eastern and central areas have successively changed from the loss of land resource allocation due to the *Non-agricultural Land Conversion* to the loss of land resource allocation efficiency due to the Agricultural Land Use. The survey data analysis indicates that the opportunity cost of agricultural labor will increase in the next 30 years due to the constraints of the continuous expansion and transfer of agricultural labor [5, 6]. It determines that agricultural land will be idle for a long time, thereby severely challenging China's food security.

Several researchers have showed that the efficiency of land resource allocation had been improved in the early stage of rural reform because of the comprehensive promotion of the household contract system [7, 8]. The rapid growth of grain output manifested this concept. With the release of internal incentives of farmland property rights, the production efficiency of farmland resources decreased slowly after 1984 [9]. The instability of farmland property rights derived from the problem is evident. On the one hand, unstable property rights indicate that farmers cannot enjoy full exclusive rights to control agricultural land, reducing the willingness to invest in agricultural production factors and inducing the abandonment of inefficient land. On the other hand, unstable farmland property rights inhibit the implementation of farmland transfer right and increase the cost of recovering farmland transfer right. Thus, the possibility of farmers transferring abandoned land is reduced, and finally agricultural land resource allocation decreases [10]. The general view is that since the implementation of the household contract responsibility system, the Chinese government has considered stabilizing the land contract relationship as the critical policy key but has not established a complete registration and management system for land contract and management rights. The fact that large and small adjustments are evident during the contract period results in unstable farmland property rights [10–12]. On the basis of actual measurement, the right of land contract and management of peasant households is defined in the form of a property right certificate. In essence, it is expressed as follows: the exclusive highest right of land control is granted to peasant families. Therefore, stable property rights improve the agricultural land resource allocation and output efficiency and objectively reduce the farmland wastage behavior of farmers [13, 14].

The contrast between the conclusions of the existing literature and the actual situation is vital. China's farmland property rights constantly improve, with the policy motive of *empowering and increasing energy*. The promotion of a new round of farmland ownership confirmation continued from the first pilot to the gradual expansion of the pilot scope [15, 16]. From 2009 to 2010, the Ministry of Agriculture, for the first time, confirmed the pilot work of eight villages in eight provinces, including Sichuan Province and explored the promotion of the entire town. From 2011 to 2013, towns were considered units. In 2013, the work expanded to 105 counties (cities and districts), including the Pinggu District of Beijing. In 2014, the county was considered a unit, and the entire province pilot in Shandong, Sichuan and Anhui was explored. In 2015, the scope of the pilot program continued to expand, and nine new pilot provinces (regions) in Jiangsu, Jiangxi, Hubei, Hunan, Gansu, Ningxia, Jilin, Guizhou, and Henan were added. However, according to incomplete statistics, China's arable land area decreased by 1.44 million hm2 from 2000 to 2015 alone. Nearly 85% of agricultural land is caused by farmers' abandonment [17, 18]. Therefore, the farmers' farmland use behavior does not show the occurrence logic of linear response to farmland ownership confirmation and certification, and

the impact of farmland ownership confirmation on farmers' wastage behavior is situational dependent.

A noteworthy judgment is that the property right allocation of the household contract responsibility system is similar to the principal-agent mechanism. The farmers' land wastage behavior is equivalent to unethical behavior in the principal-agent relationship. Therefore, the residual control right and residual claim implicit in accurate farmland affirmation can motivate farmers to maximize agricultural production income. However, the realization of residual control by farmers is related to the surplus value of agrarian production. Farmers with considerable surplus value of agricultural output can reduce the motivation of farmland wastage by expanding production scale under the incentive of a residual claim. On the contrary, the farmers' surplus value of agricultural production is small. Thus, they cannot easily pursue the goal of increasing production and income with the incentive of the residual claim. As residual control reduces the risk of land loss, farmers are more inclined to transfer the labor force off-farm, ultimately manifested as farmland wastage behavior. On this basis, the data from the 2016 CLDS survey of Sun Yat-sen University and in accordance with the principal-agent mechanism of the household contract system are adopted to investigate the effects of the implicit residual rights of control and residual claims on the moral behavior of farmers in the farmland abandonment. The results show that the impact of farmland right confirmation on farmers' farmland abandonment behavior is situation dependent, and the change in situation and constraint conditions induces the transformation of farmers' farmland abandonment decision. Furthermore, the implementation effect of land ownership confirmation and certification is not an evident result, with gradual and internal imbalance.

## 2. Theoretical analysis

**2.1 Commission-agent relationship: Collective ownership of farmland and contracted management by peasant households.**   The household contract responsibility system is a type of "collective ownership" agricultural land system. The existing laws and regulations clarify that the land contracted by peasant households belongs to the collective ownership of the villagers' groups to which the peasant households belong. As the ownership subject, the collective owns the rights of distribution, adjustment, and recovery of agricultural land [19, 20]. However, prior to promulgation of the General Provisions of the Civil Law in 2017, the collectives did not legally have extraordinary legal personality and were not natural and legal persons. Therefore, the direct joint exercise of ownership of the right of possession faces substantial transaction costs. Therefore, the separation of ownership and control rights can be realized in the "principal-agent" mechanism, effectively reducing the system cost of effective operation of farmland property rights [21].

The household contract responsibility system embodies the compound property of principal-agent in the arrangement of property rights contract. On the one hand, the collective land ownership in China's rural areas was formed when farmers, who joined the high-level cooperatives in 1957, surrendered their lands and transformed them into joint ownership by cooperatives [22]. Collectively owned lands are derived from land ceded by farmers in the same year. Thus, all the farmers "entrust" the farmland management right of the village to the village collective organization, forming the primary principal-agent relationship, with all the farmers as the initial principal and the town collaborative as the agent [23]. Paragraph 2 of Article 10 of the Constitution and Articles 58, 59, and 60 of the Property Law stipulate that the ownership of the collective land of farmers belongs to the collective members of the collective.

Since 1978, the rural collective land has been under the household contract responsibility system of contracting all land production to each household. Farmers and collaborative

economic organizations sign land contract management contracts and obtain land contract management rights, forming a secondary principal-agent relationship, with the village collective as the principal and peasant household as the agent, indicating the principal-agent relationship of land contract management rights. Articles 5 and 21 of the Rural Land Contracting Law stipulate that members of rural collective economic organizations shall have the right to contract rural land to be contracted by their collective economic organizations in accordance with the law and the contracting party shall sign a written contract with the contractor. Compared with the entrustment and agency of ownership in the 1950s, the entrustment and agency of land contracting and management rights, which has been the basic system of rural collective land since the 1980s, have a more profound impact on the rural land use behavior of farmers.

## 2.2 The unethical behavior under principal-agent mechanism: The behavior of farmers throwing away land

The unethical behavior of peasant households in the utilization of farmland is manifested as the situation of peasant households seeking the optimal input and output. Farmers' land wastage behavior stems from the failure of the land property right system to form incentive effect with constraint boundary because of resource mismatch due to the loss of production efficiency. Prior to the new round of farmland ownership confirmation, farmland property rights regarding spatial attributes and property rights attributes are vague [24]. Moreover, the genetic logic of moral failure behavior was implied, and motivating farmers to form the consciousness of intensive and economical utilization of all plots was difficult. The main effect of collective ownership of farmland on the cooperative members of peasant households is to satisfy the survival and social security requirements of collective members through producing farmland attributes [25]. The rationality of joint boundary adjustment of contracted management rights is determined in accordance with the man-land relationship. On the basis of "rural land contract law amendment" and the 20th regulation, "the contract period of arable land is 30 years." Therefore, if the land contract management right can be adjusted or recovered by collective land ownership, then the land contract management right owned by peasant households is temporary but equivalent to the usufruct right with a longer term [26]. In fact, as the legal owners of farmland, the village collective and even the village cadres can take advantage of the dominant agricultural land adjustment, add property right preference and private interests [27], and then benefit from the efforts of farmers to improve their land productivity. From this point of view, the actions of farmers to improve land fertility have positive externalities. In the unstable situation of land right confirmation, farmers produce two types of behavior logic of moral defeat. First, they produce a large amount of fertilizer and medicine and tend to shorten agricultural production to thoroughly plunder the surplus value of the farm output. Second, they abandon low production efficiency, even general plots of land and reduce the risk of agricultural production investment being "free riders." In addition, to consider efficiency and fairness, the household contract responsibility system, whether "distribution according to man" or "distribution according to work," requires the allocation of land rights according to land quality, distance, and irrigation conditions, thereby forming a "good and bad match" of multi-equal partition pattern. Farmers can use agricultural land in different places to plunder the land. In addition, the efficiency of farm production and management is unobservable; thus, the efforts of farmers to improve land fertility are often measured through agricultural output. As the basic management unit of agricultural production, the farm output of peasant households is concealed. Thus, all circles of society cannot effectively judge peasant households' moral failure behavior. Furthermore, farmers use considerable amount of chemical fertilizers and pesticides in the plots with good quality to improve the production efficiency of the

farmland, and in plots with poor quality, the agricultural investment for the improvement of soil fertility is reduced; it is finally manifested as the phenomenon of idle farmland.

The "principal-agent" arrangement contract of the household contract responsibility system has intensified the effect of property instability on farmers' farmland wastage behavior. The existing laws and regulations indicate that the rural collective land belongs to the collective ownership of township farmers, village farmers, and villagers' group farmers [28]. The subject of joint ownership is vague. Although Articles 13 and 60 of the Rural Land Contracting Law stipulate the following: "to supervise the contractor to reasonably use and protect the land according to the purposes agreed in the contract"; "if the contractor causes permanent damage to the contracted land, then the employer has the right to stop it and demand compensation for the losses from the contractor." However, in the actual operation process, the collective ownership subject's power in the management of agricultural land is empty, and detailed rules for guidance and financial support are limited. As a result, the farmland property right is wholly exposed in the public domain, and restricting the farmland abandonment behavior of farmers is difficult. Meanwhile, Article 5 of the Law on Rural Land Contracting stipulates that "members of rural collective economic organizations have the right to contract rural land by their collective economic organizations according to law." Therefore, the peasant households' land contract management right is based on collective ownership and will not be eliminated once realized. Thus, as long as joint land ownership and peasant households' collective membership exist, peasant households can obtain the right to contract out communal land. The persistence of personality rights due to identity attributes determines the difficulty of restraining farmers' desolation behavior through withdrawal rights, thereby exacerbating the deviation of land resource allocation from the Pareto optimal state.

## 2.3 The imbalance of incentives: Impact of land ownership confirmation on farmers' decision to abandon the land

In 1984, the Chinese government clarified that the contracted land-use rights would remain the same for 15 years. In 1998, the acquired land-use rights were extended for 30 years, and land contracts or certificates were issued to farmers accordingly [29]. However, the actual issuance of the certificate of right is relatively low, indicating that the unstable and incomplete property rights of agricultural land are widespread. For this reason, the Ministry of Agriculture issued the Opinions on the Pilot Work of Registration of Contracted Management Rights of Rural Land in 2011. It stipulates that "on the basis of the actual measurement of land, the farmers' contracted land should be confirmed, registered, and certified"; the contracted land, area, contract, and ownership certificate can be fully implemented to the households [30].

In the principal-agent mechanism of land contract and management right, the confirmation of farmland right is considered an incentive contract that "provides farmers with the internalization of external benefits" [31]. Accurate affirmation of rural land solidifies farmers' residual control right and residual claim on agricultural land, and then changes the efficiency of farmers' allocation of agricultural land resources [32]. Among them, residual control rights refer to the rights that have not been defined in the contract. Residual rights of control determine that the subject of the property right "may exercise any right in a way that is consistent with the prior contract, custom, or law" [33]. Land ownership confirmation endows farmers with the decision-making right of matters not specified in the land contract and management cooperation, thereby improving the security and stability of farmland property rights and providing farmers a free space to dispose the farmland and allocate production factors [34]. The residual control right implicit in the accurate farmland affirmation guarantees the farmers' right of free use of farmland (including farmers' wastage behavior) without being "occupied"

by a third party. The residual claim refers to the "claim" of supervisors on the net income generated by their additional investment [35]. The size of the surplus is related to the subject of property rights [36]. Therefore, the farmland right confirmation solidifies the land contract management right in the property right certificate. It defines how farmers benefit and lose in economic activities, thereby helping farmers to form reasonable profit and loss expectations in the agricultural production and operation market. Logically, the residual claim implicit in the confirmation of agricultural land rights can act on the farmers' consciousness of pursuing excess interests in agricultural production and stimulate farmers' rational allocation of land resources to the greatest extent.

Consequently, Residual control rights guarantee the free space of farmers' farmland utilization. And residual claim stimulates the goal of maximizing the benefit of agricultural land resource allocation. The residual claim is the key factor to change the decision of farmers on farmland wastage. Further, to obtain the greater surplus value of agricultural production, farmers become more willing to plant cash crops or intertemporal crops, resulting in more incentives for investment in agricultural production factors and improving the land quality problem caused by "good and bad combination" of contracted land [37]. Farmers can obtain the excess value of agricultural production through agricultural production because the quality of leased land has been dramatically improved, encouraging farmers to optimize the allocation of agricultural land resources and objectively reducing the phenomenon of farmland wastage (e.g, S1 Fig).

Although residual claims are expressed essentially as incentives to claim or create assets, they are related to the constraint conditions of farmers' agricultural production. The reduction of farmland wastage by farmers indicates reproduction on a larger scale. Its foundation is to use part of the surplus value for productive accumulation. Therefore, the incentive of residual control rights brought by accurate farmland affirmation is different for low-income families and nonpoor families (e.g, S2 Fig). On the premise of the same labor factors, nonpoor families tend to have more evident capital and technology factors than low-income families. Nonpoor households are more likely to obtain more surplus value during agricultural production. Therefore, nonpoor families pursue excess profits in agricultural production, whereas low-income families may consider "satisfying basic survival needs" as the production goal. Nonpoor households can transform the surplus value of agricultural production into expanding the capital factors of agricultural reproduction. They can improve production efficiency by allocating agricultural production factors, thereby reducing the farmland wastage behavior.

On the contrary, the surplus value generated by poor households in the agricultural production process is lower. They mainly use the surplus value for necessary consumption materials rather than into the physical form of survival materials. Poor peasant households cannot easily pursue excess profit in agricultural production due to the restriction of agricultural production conditions in realizing surplus claim in agricultural production. On the premise of increasing external employment space, they are more willing to work and receive labor remuneration from nonagricultural industries due to the low effect of agricultural production on the disposable income of poor household farmers. In particular, land ownership confirmation strengthens farmers' residual control over farmland and reduces the risk of land loss that farmers may face when they engage in off-farm employment; it is further reflected in the reduction of off-farm labor transfer costs. As a result, poor households are more likely to move out of farming altogether and tend to sideline or abandon farming for long periods. Moreover, the rational allocation of residual control rights can produce additional residual income, and farmers prefer to retain residual control rights. Compared with the transfer of farmland, controlling the farmland by leaving the farmland idle is more advantage for poor household farmers. The farmland transfer contract is incomplete; thus, it can induce the opportunistic behavior of

farmland flowing into households to lose the land. The primary concern of low-income family farmers is not the economic return brought by farmland transfer but the transaction cost of recovering farmland intact and the land after the lease expires, considering the risk of future unemployment and retirement. From this point of view, poor farmers pursue the labor remuneration of nonagricultural industries and the "residual claim" of farmland as the risk carrier of social security; it is ultimately manifested as the idling of farmland and the improvement efforts of land fertility similar to "fallow."

Hypothesis 1: For low-income families, farmland ownership affirmation stimulates farmers' awareness of pursuing nonagricultural labor remuneration, promoting farmland wastage behavior.

Hypothesis 2: For nonpoor households, land ownership affirmation stimulates farmer awareness of pursuing the right to claim the surplus of agricultural land production, thereby reducing farmland wastage behavior.

## 3. Data source and variable selection

### 3.1 Data sources and descriptive statistics

The data for this study are obtained from the 2016 China Labor Force Dynamics Survey (CLDS) conducted by the Social Science Survey Center of Sun Yat-sen University. The CLDS was granted ethical approval from the Research Ethics Committees of the Social Science Research Center of Sun Yat-sen University. All subjects signed written informed consent before the interview. The survey uses a multistage, multilevel sampling method proportional to the size of the labor force and randomly samples 29 provinces across the country through rotating sample tracking. The research data include three levels, namely, individual, family, and village data. After removing outliers, invalid answers, and other samples, data of 14,226 respondents were obtained.

In Table 1, the different degrees of family characteristics and village characteristics between households with confirmed and nonconfirmed rights are reported on the unit of households. The results showed almost no difference in family characteristics between the households with and without rights. Among the characteristics of villages, except for the slight differences in the proportion of village engaged in agriculture and the proportion of labor over 60 years old,

**Table 1. Farmland confirmation and farmers' family and village characteristics.**

| Constructs | | Whole sample (1) | Households with the rights (2) | Households without the rights (3) | Diff (4) = (2)—(3) |
|---|---|---|---|---|---|
| Family characteristics | Waste area of cultivated land | 0.385 (2.964) | 0.389 (2.372) | 0.393 (3.680) | -0.004 |
| | Engel coefficient | 0.426 (0.252) | 0.382 (0.248) | 0.402 (0.252) | -0.020 |
| | Demolition experienced | 0.008 (0.087) | 0.007 (0.083) | 0.008 (0.092) | -0.001 |
| | Ratio of agricultural income to household income | 0.611 (0.402) | 0.626 (0.399) | 0.588 (0.410) | 0.038 |
| | Area of cultivated land | 6.439 (49.480) | 8.780 (69.774) | 4.46 (22.691) | 4.32 |
| Village characteristics | Ratio of agricultural work | 42.649 (40.947) | 67.713 (32.959) | 57.908 (37.656) | 9.805 |
| | Number of seasonal migrant workers | 548.949 (5420.321) | 255.416 (291.814) | 256.887 (353.738) | -1.471 |
| | Ratio of labor force over 60 years old | 22.801 (25.759) | 33.338 (25.139) | 29.649 (25.698) | 3.689 |
| | Land adjustment | 0.256 (0.702) | 0.201 (0.544) | 0.308 (0.835) | -0.107 |
| | Idle land | 0.262 (0.440) | 0.179 (0.384) | 0.347 (0.476) | -0.168 |
| | Soil transformation | 0.135 (0.342) | 0.141 (0.348) | 0.126 (0.332) | 0.015 |
| | Forest rehabilitation from slope agriculture | 0.395 (0.489) | 0.449 (0.498) | 0.341 (0.474) | 0.108 |

the differences in other aspects are unnoticeable. This finding indicates that the selection of pilot projects for the determination and certification of farmland rights should be based on the principle of representativeness and universality. In exploring the influence of accurate affirmation on farmland wastage, *selective* right affirmation does not cause serious endogeneity problems.

## 3.2 Variable selection

**3.2.1. Explained variable.** "Whether farmers abandon farmland (1 = yes; 0 = No)". In the robustness test, "the area of farmland abandoned by farmers" and "the area of paddy field or irrigated land abandoned by farmers" are selected as the measure items of explained variables. Table 2 shows the specific variable selection.

**3.2.2. Explanatory variable.** The measured item of the explanatory variable was selected as "whether farmers receive the Certificate of Contracted Operation of Rural Land."

**3.2.3. Control variables.** We have included four levels of control variables in this paper. First, gender, age, and marital status are selected as the measure of personal characteristics. Among them, women face less off-farm employment opportunities than men, and older farmers or married farmers face greater competition in the off-farm employment market. They may rely on agricultural employment security and thus be constrained by the land tenure confirmation to reduce farmland wastage.

**Table 2. Variable description and descriptive statistics analysis.**

| Variable | | Variable assignment | Mean | Standard deviation |
|---|---|---|---|---|
| Explained variable | Farmland abandonment behavior | whether farmers abandon farmland (1 = Yes; 0 = No) | 0.117 | 0.322 |
| | Waste area of cultivated land | Actual value (mu) | 0.364 | 2.519 |
| | Irrigated land abandoned | Actual value (mu) | 0.153 | 1.033 |
| Explanatory variable | Farmland Confirmation | Whether farmers receive the Certificate of Contracted Operation of Rural Land | 0.510 | 0.500 |
| Personal characteristics | Gender | 1 = Male; 2 = Female | 1.525 | 0.499 |
| | Age | Actual value (years) | 43.815 | 14.612 |
| | Marital status | 0 = Unmarried; 1 = Married; 2 = Divorced; 3 = Widowed | 1.912 | 0.501 |
| Family characteristics | Family size | Actual number of family | 4.432 | 2.091 |
| | Agricultural production and operation cost | Actual value (Yuan) | 5914.384 | 10501.180 |
| | Total household income | Actual value (Yuan) | 60183.060 | 94217.570 |
| | Total agricultural income | Actual value (Yuan) | 5659.922 | 12661.680 |
| Farmland characteristics | The total area of cultivated land | Actual value (mu) | 7.747 | 78.647 |
| | The area of cultivated land leased or farmed for substitution | Actual value (mu) | 15.677 | 1180.289 |
| | The area of paddy field leased or farmed for substitution | Actual value (mu) | 0.592 | 7.087 |
| Village characteristics | Nonagricultural economic opportunities | If the village has a nonagricultural economy (1 = Yes; 0 = No) | 0.265 | 0.442 |
| | Village topography | 1 = Plains; 2 = Hills; 3 = Mountains | 1.602 | 0.806 |
| | Village traffic | Percentage of the traffic road that is hardened pavement (%) | 71.748 | 24.886 |
| | Conversion of farmland to forest | If the village has implemented the conversion of farmland to forest (1 = Yes;0 = No) | 0.385 | 0.487 |
| | Soil reconstruction | If the village has implemented soil improvement (1 = Yes;0 = No) | 0.137 | 0.344 |
| | Proportion of cultivated land abandoned | Proportion of the area of abandoned farming and abandoned land in the village to the area of the village's land (%) | 13.914 | 13.731 |

Second, the family size, agricultural production, operation cost, total household income, and total agricultural income are selected to measure family characteristics. A larger family size is more likely to rely on agricultural employment security. The cost of agricultural production and operation, the total income of the family, and the total income of agriculture can reflect the surplus value of agricultural production and play an essential role in farmers' decision-making on agricultural wastage.

Third, the total area of cultivated land, the area of cultivated land leased or farmed for substitution, and the area of paddy field leased or farmed for substitution were selected as the measurement item of agricultural land characteristics. The area of cultivated land, the area of cultivated land leased or farmed for substitution, and the area of paddy field leased or farmed for substitution reflect the constraints of farmland utilization and impact on farmland wastage behavior.

Fourth, nonagricultural economic opportunities, village topography, village traffic, conversion of farmland to forest, soil reconstruction, and land abandonment area are selected as the measurement items of village characteristics.

### 3.3 Empirical model

To validate the influence of Farmland Confirmation on Farmland Abandonment, we consider the following model:

$$Y = \beta_0 + \beta_1 Farmland\ confirmation + \beta_2 Control\ variables + \varepsilon \tag{1}$$

In Eq (1), $Y$ is the explained variable, and *Farmland confirmation* is the Explanatory variable. *Control variables* is a set of control variables and $\varepsilon$ denotes the error term.

## 4. Results

### 4.1 Baseline regression

Model 3–1 in Table 3 reports the influence of farmland right confirmation on farmers' farmland wasteland behavior in the entire sample. The results show that the coefficient of farmland right confirmation is positive and insignificant. This finding indicates that the impact of farmland right confirmation on farmers' farmland wastage behavior is uncertain and situational dependent. Thus, the implementation effect of land ownership confirmation and certification is nonapparent; a gradual and internal imbalance is observed. Model 3–2 reports the impact of land ownership confirmation on farmers' farmland wastage behavior in the sample of poor households. The results show that at the statistical level of 1%, the coefficient of farmland right affirmation is significantly positive. This finding indicates that, for poor households, land ownership affirmation has a promoting effect on farmland wastage behavior, thereby confirming Hypothesis 1. Model 3–3 reports the impact of land ownership confirmation on farmers' farmland wasteland behavior sample of nonpoor households. The results show that at the statistical level of 5%, the coefficient of farmland right confirmation is significantly harmful. It indicates that, for farmers from nonpoor families, land ownership affirmation has an inhibitory effect on their wastage behavior, thereby confirming Hypothesis 2.

The results in Table 3 reveal that the accurate farmland affirmation essentially solidifies the farmers' residual control right and residual claim right on farmland. Therefore, farmers can pursue the value of agricultural land or maximize the income of agricultural production within the "principal-agent" mechanism of the household contract responsibility system. The behavior of farmers' land use will undoubtedly be differentiated in two aspects. Farmers become "resource-based farmers," who tend to transfer their farmland to leading enterprises.

Table 3. Impact of farmland confirmation on farmers' behavior of farmland waste.

| Variable | Model 3–1: Entire sample | | Model 3–2: Poor peasant households | | Model 3–3: Nonpoor peasant households | |
|---|---|---|---|---|---|---|
| | Parameter estimate | Marginal estimate | Parameter estimate | Marginal estimate | Parameter estimate | Marginal estimate |
| Farmland Confirmation | 0.038 (0.113) | -0.048 (0.023) | 1.009*** (0.412) | 0.133*** (0.051) | -0.383** (0.190) | -0.048** (0.023) |
| Gender | 0.087 (0.108) | 0.026 (0.022) | -0.183 (0.325) | -0.024 (0.043) | 0.204 (0.176) | 0.026 (0.022) |
| Age | 0.006 (0.004) | 0.002 (0.001) | 0.018 (0.014) | 0.002 (0.002) | 0.014** (0.007) | 0.002** (0.001) |
| Marital status | -0.018 (0.128) | -0.004 (0.025) | -0.110 (0.284) | -0.015 (0.037) | -0.030 (0.203) | -0.004 (0.025) |
| Family size | 0.113*** (0.024) | 0.010*** (0.005) | -0.010 (0.119) | -0.001 (0.016) | 0.079** (0.040) | 0.010 (0.005) |
| The total area of cultivated land | 0.058*** (0.008) | 0.013*** (0.002) | 0.000 (0.032) | 0.000 (0.004) | 0.101*** (0.015) | 0.013*** (0.002) |
| The area of cultivated land leased or farmed for substitution | -0.106*** (0.035) | -0.017 (0.007) | -0.202*** (0.076) | -0.027*** (0.011) | -0.138*** (0.053) | -0.017*** (0.007) |
| The area of paddy field leased or farmed for substitution | 0.153*** (0.039) | 0.032*** (0.007) | 0.380*** (0.159) | 0.050*** (0.022) | 0.255*** (0.061) | 0.032*** (0.007) |
| Agricultural production and operation cost | 0.000 (0.000) | 0.000 (0.000) | 0.000 (0.000) | 0.000 (0.000) | 0.000 (0.000) | 0.000 (0.000) |
| Total household income | 0.000 (0.000) | 0.000 (0.000) | 0.000 (0.000) | 0.000 (0.000) | 0.000** (0.000) | 0.000** (0.000) |
| Total agricultural income | 0.000 (0.000) | 0.000 (0.000) | 0.000 (0.000) | 0.000 (0.000) | 0.000** (0.000) | 0.000** (0.000) |
| Non-agricultural economic opportunities | -0.138 (0.133) | -0.010 (0.026) | -1.309*** (0.402) | -0.173*** (0.052) | -0.077 (0.208) | -0.010 (0.026) |
| Village topography | -0.098 (0.071) | -0.017 (0.015) | -0.627*** (0.233) | -0.083*** (0.030) | -0.137 (0.123) | -0.017 (0.015) |
| Village traffic | -0.003 (0.002) | 0.001 (0.000) | 0.003 (0.007) | 0.000 (0.001) | 0.005 (0.003) | 0.001 (0.000) |
| Conversion of farmland to forest | -0.183 (0.119) | -0.105 (0.024) | -0.967*** (0.383) | -0.127*** (0.048) | -0.839*** (0.193) | -0.105*** (0.024) |
| Soil reconstruction | -0.223 (0.166) | -0.012 (0.034) | 0.080 (0.416) | 0.011 (0.055) | -0.093 (0.271) | -0.012 (0.034) |
| Proportion of cultivated land abandoned | 0.017*** (0.004) | 0.004*** (0.001) | 0.027*** (0.013) | 0.004*** (0.002) | 0.030*** (0.006) | 0.004*** (0.001) |
| Constant | -2.369*** (0.411) | - | -0.795 (1.174) | - | -3.005*** (0.691) | - |
| P-value | 0.000 | - | 0.000 | - | 0.000 | - |
| Adjusted R$^2$ | 0.057 | - | 0.149 | - | 0.148 | - |

Note

*, **, and *** represent the significance levels of 1%, 5%, and 10%, respectively; the robust standard errors are in parentheses.

Furthermore, farmers become "production-oriented farmers," who will be inclined to optimize the allocation of agricultural production factors. These results can objectively reduce the phenomenon of farmland abandonment. One notable fact is that farmers' "selfish motives," given the constraints, make the most satisfactory decisions. Therefore, the effect of the residual control right and residual claim implicit in farmland right affirmation on poor farmers and nonpoor farmers is different. For poor peasant households, the residual control right reduces the risk of land loss in their nonagricultural transfer, and the residual control right urges them to pursue the social security income of farmland; it is finally shown as the farmland abandonment behavior. For nonpoor farmers, the residual control rights guarantee the farmers' expectation of recovering agricultural investment, and the residual claim stimulates their production goal of increasing production and income, thereby reducing the occurrence of farmland wastage.

## 4.2 Robustness test

Table 4 shows the results of the robustness test, which was conducted by substituting the explained variables. Model 4–1 reports the influence of land ownership confirmation on

**Table 4. Farmland confirmation and farmland waste area of farmers.**

| Variable | Model 4–1: Waste area of cultivated land | | Model 4–2: Waste area of paddy field or irrigated land | |
|---|---|---|---|---|
| | Poor peasant households | Nonpoor peasant households | Poor peasant households | Nonpoor peasant households |
| Farmland Confirmation | 1.208** (0.542) | -0.729***(0.159) | 0.494**(0.203) | -0.778***(0.142) |
| Gender | 0.093 (0.487) | 0.010 (0.146) | -0.091 (0.182) | -0.029(0.137) |
| Age | 0.007 (0.020) | -0.004 (0.005) | -0.001 (0.008) | -0.002 (0.005) |
| Marital status | 0.076 (0.485) | -0.153 (0.136) | 0.036 (0.182) | -0.062 (0.163) |
| Family size | 0.296** (0.147) | -0.047** (0.023) | 0.145*** (0.055) | -0.022 (0.032) |
| The total area of cultivated land | 0.078** (0.041) | 0.239*** (0.041) | -0.011 (0.016) | 0.175*** (0.010) |
| The area of cultivated land leased or farmed for substitution | -0.136 (0.086) | -0.053 (0.037) | -0.054* (0.032) | -0.047*** (0.010) |
| The area of paddy field leased or farmed for substitution | 0.249 (0.241) | 0.144*** (0.040) | -0.190**(0.090) | 0.065*** (0.026) |
| Agricultural production and operation cost | 0.001*** (0.000) | 0.000 (0.000) | 0.000*** (0.000) | 0.000* (0.000) |
| Total household income | 0.000 (0.000) | 0.000 (0.000) | 0.000 (0.000) | 0.000 (0.000) |
| Total agricultural income | 0.000*** (0.000) | 0.000 (0.000) | 0.000** (0.000) | 0.000** (0.000) |
| Non-agricultural economic opportunities | 0.245 (0.611) | 0.597*** (0.165) | 0.103 (0.229) | 0.536*** (0.162) |
| Village topography | -1.117*** (0.362) | -0.313*** (0.093) | -0.349*** (0.136) | -0.176** (0.095) |
| Village traffic | -0.001 (0.010) | 0.002 (0.003) | -0.005 (0.004) | -0.003 (0.003) |
| Conversion of farmland to forest | -1.096 (0.616) | -0.552*** (0.113) | -0.414* (0.231) | -0.522*** (0.150) |
| Soil reconstruction | -0.496 (0.659) | -0.004 (0.138) | -0.156 (0.247) | -0.012 (0.193) |
| Proportion of cultivated land abandoned | 0.040** (0.023) | 0.019*** (0.005) | 0.000 (0.009) | 0.006 (0.005) |
| Constant | 0.158 (1.844) | 0.888 (0.604) | 0.788 (0.691) | 0.784 (0.552) |
| P-value | 0.000 | 0.000 | 0.000 | 0.000 |
| Adjusted $R^2$ | 0.273 | 0.401 | 0.184 | 0.315 |

Note

*, **, and *** represent the significance levels of 1%, 5%, and 10%, respectively; the robust standard errors are in parentheses.

farmland wasteland area of farmers. Model 4–2 reports the influence of land ownership confirmation on the abandoned area of farmers' paddy field or irrigated land. The results are robust; that is, land ownership confirmation has a promoting effect on the area of farmland abandoned by farmers for low-income families. For nonpoor families, land ownership confirmation has a restraining effect on the area of farmland abandoned.

## 5 Conclusions and discussions

The academic community generally believes that farmland accurate affirmation enhances the stability and security of property rights, which translate into the efficiency of agricultural land resource allocation and then reduce farmers' farmland wastage behavior. This study holds that the policy of confirming and certifying farmland rights mainly expresses the attribution mark of the land contract management rights granted to farmers by law, and is the specific description of real rights. The appropriate definition of mutual interest relationship between people caused by the implicit residual control right and residual claim in the confirmation and certification of farmland ownership is the inducing factor of property rights on farmland utilization behavior.

The research conclusions are as follows: (1) the household contract responsibility system is essentially a "principal-agent" mechanism, in which the farmers' land wastage is their unethical behavior. Prior to the implementation of a new round of farmland ownership confirmation

and certification policy, farmers are more likely to have moral hazard, and then abandon low-quality, even ordinary land due to the lack of ownership certificate to implement the residual control rights and residual claim incentive to households. (2) Rural land ownership confirmation is relative to the contract that motivates farmers to optimize the allocation of agricultural land resources. Its essence is to strengthen the farmer's residual control right and residual claim right, and then protect the farmer's free space of agricultural land utilization, thereby stimulating the pursuit of profit maximization action. (3) The realization of farmers' residual claim is related to the constraints of agricultural production. For low-income families, the surplus value of agricultural production is low. Thus, it restricts the goal of increasing production and income by expanding agricultural production. Residual control rights reduce the risk of land loss in nonagricultural transfer and eventually lead to the phenomenon of farmland wastage. For nonpoor families, the surplus value of agricultural production is higher. Farmers are more willing to increase agricultural investment to obtain surplus value, improving the agricultural land resource allocation efficiency and reducing the phenomenon of farmland wastage.

Policy implication: The result of land right confirmation and land wastage indicates that the effect of land right confirmation on farmers' land use behavior is not apparent. The effect of farmland accurate affirmation is not balanced and progressive. Therefore, the institutional basis of the matching policy of farmland ownership confirmation should be to introduce the policy according to local conditions and deal with the relationship between the implicit residual control right and residual claim right of farmland ownership confirmation.

Possible innovations: the literature on the determination of farmland rights has received less attention to residual control rights and residual claims. This study is a preliminary expansion. Although the existing literature has focused on the influence of farmland accurate affirmation on farmland wastage, it has ignored the situational dependence, which is discussed in this study. From the perspective of improving the efficiency of farmland resource allocation, a possible explanatory framework is provided for enhancing the incentive effect of farmland right confirmation on farmers' farmland use behavior.

## Supporting information

**S1 Fig. Impact of farmland confirmation on farmers' abandonment behavior.**
(TIF)

**S2 Fig. Residual control right, residual claim right, and farmers' abandonment decision.**
(TIF)

**S1 File. CLDS2016-questionnaire.**
(DOC)

**S2 File. The data in this research originated from the 2016 China Labor-Force Dynamic Survey (CLDS).**
(RAR)

## Author Contributions

**Conceptualization:** Zhidong Wu.

**Data curation:** Zechen Yang.

**Writing – original draft:** Wolin Zheng.

**Writing – review & editing:** Wolin Zheng.

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
