## [Decision Letter · Decision Letter 0]

8 Dec 2022

PONE-D-22-26169Influence of Farmland Confirmation on Farmland Abandonment in ChinaInfluence of Farmland Confirmation on Farmland Abandonment in ChinaPLOS ONE

Dear Dr. Wolin Zheng,

Thank you for submitting your manuscript to PLOS ONE. After careful consideration, we feel that it has merit but does not fully meet PLOS ONE’s publication criteria as it currently stands. Therefore, we invite you to submit a revised version of the manuscript that addresses the points raised during the review process.

We look forward to receiving your revised manuscript.

Kind regards,

Bo Huang

Academic Editor

PLOS ONE

Journal Requirements:

2. Please ensure that you have specified (1) whether consent was informed and (2) what type you obtained (for instance, written or verbal, and if verbal, how it was documented and witnessed). If your study included minors, state whether you obtained consent from parents or guardians. If the need for consent was waived by the ethics committee, please include this information.

3. Our internal editors have looked over your manuscript and determined that it is within the scope of our Sustainability and the Circular Economy Call for Papers. The Collection will encompass a diverse and interdisciplinary set of submissions related to sustainability and the circular economy, focusing on production models, business plans, and the contribution of global initiatives to increased sustainability in economic, environmental, and social terms. Additional information can be found on our announcement page: https://collections.plos.org/call-for-papers/sustainability-and-the-circular-economy/. If you would like your manuscript to be considered for this collection, please let us know in your cover letter and we will ensure that your paper is treated as if you were responding to this call. If you would prefer to remove your manuscript from collection consideration, please specify this in the cover letter.

” The work was supported by National Social Science Foundation of China (No.20&ZD170); and Academic year 2019-2020 Doctoral Dissertation Scholarship of Tsinghua Rural Studies (No.201901)；and Qingyuan City Philosophy and Social Science Planning Project for 2022: "Research on Eco-logical Asset Assessment and its Value Realization Path in Qingyuan City" (QYSK2022018)；and Research on Social Responsibility Development of Private Enterprises in Foshan and the Motiva-tion Mechanism of Participating in Rural Revitalization" (2022-GJ056).”

“NO”

6. PLOS requires an ORCID iD for the corresponding author in Editorial Manager on papers submitted after December 6th, 2016. Please ensure that you have an ORCID iD and that it is validated in Editorial Manager. To do this, go to ‘Update my Information’ (in the upper left-hand corner of the main menu), and click on the Fetch/Validate link next to the ORCID field. This will take you to the ORCID site and allow you to create a new iD or authenticate a pre-existing iD in Editorial Manager. Please see the following video for instructions on linking an ORCID iD to your Editorial Manager account: https://www.youtube.com/watch?v=_xcclfuvtxQ.

7. Thank you for stating the following in the Acknowledgments Section of your manuscript:

“The work was supported by National Social Science Foundation of China (No.20&ZD170); and Academic year 2019-2020 Doctoral Dissertation Scholarship of Tsinghua Rural Studies (No.201901).”

“The work was supported by National Social Science Foundation of China (No.20&ZD170); and Academic year 2019-2020 Doctoral Dissertation Scholarship of Tsinghua Rural Studies (No.201901)；and Qingyuan City Philosophy and Social Science Planning Project for 2022: "Research on Eco-logical Asset Assessment and its Value Realization Path in Qingyuan City" (QYSK2022018)；and Research on Social Responsibility Development of Private Enterprises in Foshan and the Motiva-tion Mechanism of Participating in Rural Revitalization" (2022-GJ056).”

8. Please ensure that you refer to Figure 2 in your text as, if accepted, production will need this reference to link the reader to the figure

Reviewers' comments:

Reviewer's Responses to Questions

**Comments to the Author**

1. Is the manuscript technically sound, and do the data support the conclusions?

Reviewer #1: Partly

Reviewer #2: Yes

2. Has the statistical analysis been performed appropriately and rigorously? 

Reviewer #1: Yes

Reviewer #2: Yes

3. Have the authors made all data underlying the findings in their manuscript fully available?

Reviewer #1: No

Reviewer #2: Yes

4. Is the manuscript presented in an intelligible fashion and written in standard English?

Reviewer #1: No

Reviewer #2: No

5. Review Comments to the Author

Reviewer #1: 1. The explained variable uses "whether farmers give up farmland, only using yes or no, but some people have not completely given up farmland. Whether we are considering whether we can use several options to distinguish intensity, which can more accurately reflect farmers' attitude to farmland. Even if there is a robust test, the measurement of this variable is not appropriate.

2. There is no econometric model in the paper.

Reviewer #2: Very good topic and timely to the moment and well presented. However, there are minor things to consider for the paper to be ready for publication. Please kindly read the authors guide and appropriately correct the minor grammatical errors in the paper in order to make it more suitable to the journal. And also consider sections titles and numbering. Also, the authors need to present the analytical framework in detail supported by literature, define and describe the variables used in the study and explain the model employed.

6. PLOS authors have the option to publish the peer review history of their article (what does this mean?). If published, this will include your full peer review and any attached files.

Reviewer #1: No

Reviewer #2: **Yes: **Essiagnon John-Philippe Alavo

---

## [Author Response · Author response to Decision Letter 0]

7 Mar 2023

Dear Editors, Dr. Essiagnon John-Philippe Alavo and Reviewers:

Thank you for your letter and for the reviewers’ comments concerning our manuscript entitled "Influence of Farmland Confirmation on Farmland Abandonment in China" (ID: PONE-D-22-26169). Those comments are all valuable and very helpful for revising and improving our paper, as well as the important guiding significance to our researches. We have studied comments carefully and have made correction which we hope meet with approval. We upload two manuscript, one of which is a separate file labeled 'Revised Manuscript with Track Changes' and the other one is a separate file labeled 'Manuscript'. Revised portion are marked in red in the paper. The main corrections in the paper and the responds to the comments are as following: 

1. Your ethics statement should only appear in the Methods section of your manuscript. If your ethics statement is written in any section besides the Methods, please delete it from any other section.

We have checked our intake and removed the ethics statement from other locations.

2. Please ensure that you have specified (1) whether consent was informed and (2) what type you obtained (for instance, written or verbal, and if verbal, how it was documented and witnessed). If your study included minors, state whether you obtained consent from parents or guardians. If the need for consent was waived by the ethics committee, please include this information. Your ethics statement should only appear in the Methods section of your manuscript. If your ethics statement is written in any section besides the Methods, please delete it from any other section.

Data Availability Statement: China Labor Force Dynamic Survey (CLDS) is a publicly available dataset. When applying for the right to use the data, the data user should provide true personal information to the Social Science Research Center of Sun Yat-sen University (hereinafter referred to as the “Data Provider”) and submit the completed “Basic Information of Data User Applicant Form” to the “Data Provider” via csyjzxsys@163.com, by fax or by mail to the “Data Provider”.

Ethics statement: Our research does not involve human participants.

We carefully read the PLOS ONE style templates and edit our manuscript according to the guidelines for main body, affiliations and file naming to ensure our revised manuscript meets PLOS ONE's style requirements.

4.Thank you for stating in your Funding Statement:Please provide an amended statement that declares *all* the funding or sources of support (whether external or internal to your organization) received during this study, as detailed online in our guide for authors at http://journals.plos.org/plosone/s/submit-now. Please also include the statement “There was no additional external funding received for this study.” in your updated Funding Statement.

We carefully read a lot of articles from PLOS ONE and modified our funding statement. 

“Funding: This work was supported by National Social Science Foundation of China (No.20&ZD170); and the Major projects of National Social Science Found of China (21&ZD090)；and Academic year 2019-2020 Doctoral Dissertation Scholarship of Tsinghua Rural Studies (No.201901).The funders had no role in study design, data collection and analysis, decision to publish, or preparation of the manuscript.”

5. Thank you for stating the following in your Competing Interests section: Please complete your Competing Interests on the online submission form to state any Competing Interests. If you have no competing interests, please state "The authors have declared that no competing interests exist.", as detailed online in our guide for authors at http://journals.plos.org/plosone/s/submit-now

We carefully read a lot of articles from PLOS ONE and modified our competing interests. 

“Competing interests: The authors have declared that no competing interests exist.”

6. PLOS requires an ORCID iD for the corresponding author in Editorial Manager on papers submitted after December 6th, 2016. Please ensure that you have an ORCID iD and that it is validated in Editorial Manager. 

We have validated the ORCID ID in Editorial Manager.

7. Thank you for stating the following in the Acknowledgments Section of your manuscript:We note that you have provided additional information within the Acknowledgements Section that is not currently declared in your Funding Statement. Please note that funding information should not appear in the Acknowledgments section or other areas of your manuscript. We will only publish funding information present in the Funding Statement section of the online submission form.

We have removed the funding-related text from the manuscript and update our Funding Statement in our cover letter. 

“Funding: This work was supported by National Social Science Foundation of China (No.20&ZD170); and the Major projects of National Social Science Found of China (21&ZD090)；and Academic year 2019-2020 Doctoral Dissertation Scholarship of Tsinghua Rural Studies (No.201901).The funders had no role in study design, data collection and analysis, decision to publish, or preparation of the manuscript.”

8. Please ensure that you refer to Figure 2 in your text as, if accepted, production will need this reference to link the reader to the figure.

We are sorry for the omission of Figure 2 in the article and we have clarified the placement of in our text.

9. We note that several of your files are duplicated on your submission. Please remove any unnecessary or old files from your revision, and make sure that only those relevant to the current version of the manuscript are included.

We are sorry for this mistake and correct it in our submission.

10. Please ensure that you have specified (1) whether consent was informed and (2) what type you obtained (for instance, written or verbal, and if verbal, how it was documented and witnessed). If your study included minors, state whether you obtained consent from parents or guardians. If the need for consent was waived by the ethics committee, please include this information.

Informed consent: We used the data of the China Labor-Force Dynamic Survey (CLDS). The CLDS was granted ethical approval from the Research Ethics Committees of the Social Science Research Center of Sun Yat-sen University. All subjects signed written informed consent before the interview.

11. Please ensure that the title on the details page and on the title page of the manuscript are identical.

We are ensure that the title on the details page and on the title page of the manuscript are identical.

12. Please upload a copy of all Figures which you refer to in your text. Or if the figure is no longer to be included as part of the submission please remove all reference to it within the text.

We have upload all Figures, Tables, and Files mentioned in our text as Supporting information files in Editorial Manager.

13. We note your Data Availability Statement as follows:

We have upload our underlying data as Supporting information files in Editorial Manager and update our Data Availability Statement.

Data Availability Statement: China Labor-Force Dynamic Survey (CLDS) is a publicly available dataset. When applying for the right to use the data, the data user should provide true personal information to the Social Science Research Center of Sun Yat-sen University (hereinafter referred to as the "Data Provider") and submit the completed "Basic Information of Data User Applicant Form" to the "Data Provider" via csyjzxsys@163.com, by fax or by mail to the "Data Provider".

Response to Reviewer #1:

Thank you for your careful and thoughtful examination of our paper. Incorporating your comments, as well as those of the Editor, and the other referee, resulted in a greatly improved manuscript. We address each of your comments below.

1. The explained variable uses "whether farmers give up farmland, only using yes or no, but some people have not completely given up farmland. Whether we are considering whether we can use several options to distinguish intensity, which can more accurately reflect farmers' attitude to farmland. Even if there is a robust test, the measurement of this variable is not appropriate.

In our explained variable "whether farmers give up farmland", the use of no means that the farmland has been abandoned. In order to differentiate the intensity, we use the specific number of farmers abandoned (Waste area of cultivated land and Irrigated land abandoned) in our robust test to differentiate the intensity and thus verify the robustness of our study.

2. There is no econometric model in the paper.

We have added to the econometric model in the paper.

To validate the influence of Farmland Confirmation on Farmland Abandonment, we consider the following model:

Y=β0+β1Farmland confirmation+β2Control variables+ɛ (1)

In Eq. (1), Y is the explained variable, and Farmland confirmation is the Explanatory variable. Control variables is a set of control variables and ɛ denotes the error term.

Response to Dr.Essiagnon John-Philippe Alavo

Thank you for your careful and thoughtful examination of our paper. Incorporating your comments, as well as those of the Editor, and the other referee, resulted in a greatly improved manuscript. We address each of your comments below.

Very good topic and timely to the moment and well presented. However, there are minor things to consider for the paper to be ready for publication. Please kindly read the authors guide and appropriately correct the minor grammatical errors in the paper in order to make it more suitable to the journal. And also consider sections titles and numbering. Also, the authors need to present the analytical framework in detail supported by literature, define and describe the variables used in the study and explain the model employed.

Firstly, we carefully read the authors guide from PLOS ONE and edit our manuscript according to the guidelines for main body to ensure our revised manuscript meets PLOS ONE's style requirements. Secondly，we read our paper carefully and corrected the minor grammatical errors in the paper. Thirdly, we carefully angled and numbered the titles of our paper. Fourthly，we present the analytical framework in detail supported by literature and provide Figure 1 and Figure 2 to help the reader understand the idea of our study. Finally, we define and describe the variables used in the study and explain the model employed.

Finally, we would like to express our gratitude to the editor and all the reviewers for the extremely helpful comments and for your guidance in the revision. We tried our best to improve the manuscript and made some changes in the manuscript. These changes will not influence the content and framework of the paper. We hope that our efforts have succeeded in allaying your concerns. We look forward to learning about your decision. And we express our thanks again to the referee for his time and efforts in reviewing our paper. Any remaining errors are our own.

Thank you and best regards.

---

## [Decision Letter · Decision Letter 1]

18 Apr 2023

Influence of Farmland Confirmation on Farmland Abandonment in China

PONE-D-22-26169R1

Dear Dr. Wolin Zheng,

We’re pleased to inform you that your manuscript has been judged scientifically suitable for publication and will be formally accepted for publication once it meets all outstanding technical requirements.

Kind regards,

Bo Huang

Academic Editor

PLOS ONE

Additional Editor Comments (optional):

Reviewers' comments:

Reviewer's Responses to Questions

**Comments to the Author**

1. If the authors have adequately addressed your comments raised in a previous round of review and you feel that this manuscript is now acceptable for publication, you may indicate that here to bypass the “Comments to the Author” section, enter your conflict of interest statement in the “Confidential to Editor” section, and submit your "Accept" recommendation.

Reviewer #2: All comments have been addressed

2. Is the manuscript technically sound, and do the data support the conclusions?

Reviewer #2: Yes

3. Has the statistical analysis been performed appropriately and rigorously? 

Reviewer #2: Yes

4. Have the authors made all data underlying the findings in their manuscript fully available?

Reviewer #2: Yes

5. Is the manuscript presented in an intelligible fashion and written in standard English?

Reviewer #2: Yes

6. Review Comments to the Author

Reviewer #2: All the comments are well addressed and the manuscript style has been improved. After minor edit of the English it can be published.

7. PLOS authors have the option to publish the peer review history of their article (what does this mean?). If published, this will include your full peer review and any attached files.

Reviewer #2: **Yes: **Essiagnon John-Philippe Alavo

---

## [Editor Report · Acceptance letter]

26 Apr 2023

PONE-D-22-26169R1 

Influence of Farmland Confirmation on Farmland Abandonment in China 

Dear Dr. Zheng:

I'm pleased to inform you that your manuscript has been deemed suitable for publication in PLOS ONE. Congratulations! Your manuscript is now with our production department. 

Kind regards, 

on behalf of

Professor Bo Huang 

Academic Editor

PLOS ONE